# Circumventing Drug Treatment? Intrinsic Lethal Effects of Polyethyleneimine (PEI)-Functionalized Nanoparticles on Glioblastoma Cells Cultured in Stem Cell Conditions

**DOI:** 10.3390/cancers13112631

**Published:** 2021-05-27

**Authors:** Neeraj Prabhakar, Joni Merisaari, Vadim Le Joncour, Markus Peurla, Didem Şen Karaman, Eudald Casals, Pirjo Laakkonen, Jukka Westermarck, Jessica M. Rosenholm

**Affiliations:** 1Pharmaceutical Sciences Laboratory, Faculty of Science and Engineering, Åbo Akademi University, 20520 Turku, Finland; nprabhak@abo.fi (N.P.); didem.sen.karaman@ikc.edu.tr (D.Ş.K.); eudaldcm@gmail.com (E.C.); 2Turku Bioscience Centre, University of Turku and Åbo Akademi University, 20520 Turku, Finland; jorome@utu.fi; 3Institute of Biomedicine, Faculty of Medicine, University of Turku, 20520 Turku, Finland; markus.peurla@utu.fi; 4Translational Cancer Medicine Research Program, Faculty of Medicine, University of Helsinki, 00014 Helsinki, Finland; vadim.lejoncour@helsinki.fi (V.L.J.); pirjo.laakkonen@helsinki.fi (P.L.); 5Department of Biomedical Engineering, Faculty of Engineering and Architecture, İzmir Kâtip Çelebi University, 35620 İzmir, Turkey; 6School of Biotechnology and Health Sciences, Wuyi University, Jiangmen 529020, China; 7Laboratory Animal Center, HiLIFE—Helsinki Institute of Life Science, University of Helsinki, 00014 Helsinki, Finland

**Keywords:** glioblastoma, brain cancer, therapy resistance, mesoporous silica nanoparticles, polyethyleneimine (PEI), proton-sponge effect

## Abstract

**Simple Summary:**

Glioblastoma (GB) is the most frequent brain cancer that is highly difficult to treat. As with many cancer types, associated cancer stem cells can act as a reservoir of cancer-initiating cells, constituting a major hurdle for successful therapy. Herein, we report on a discovery of the intrinsic capability of polyethyleneimine-functionalized nanoparticles (PEI-NPs) to selectively eradicate glioblastoma stem cells (GSCs), contrary to current drug-based approaches to target and successfully eradicate GB. Already at negligible doses, PEI-NPs, without any anticancer therapeutic, very potently killed multiple GSC lines but not GB cells without stem cell characteristics. Moreover, PEI-NPs was observed in tumors in mice after both intravenous and intranasal administration, where the latter constitute a non-invasive administration route for drug delivery to the brain. These results, in turn, suggest that PEI-NPs can successfully cross the blood-brain barrier for the eradication of GSCs even without any anticancer drug, whereas the same NP platform can also be used for drug delivery thus opening up potential to reach synergistic therapeutic effects. This highly surprising intrinsic effect of the NP system on both the mechanistic action and specificity of GSC eradication puts forward a promising novel aspect of nanoparticles in medicine.

**Abstract:**

Glioblastoma (GB) is the most frequent malignant tumor originating from the central nervous system. Despite breakthroughs in treatment modalities for other cancer types, GB remains largely irremediable due to the high degree of intratumoral heterogeneity, infiltrative growth, and intrinsic resistance towards multiple treatments. A sub-population of GB cells, glioblastoma stem cells (GSCs), act as a reservoir of cancer-initiating cells and consequently, constitute a significant challenge for successful therapy. In this study, we discovered that PEI surface-functionalized mesoporous silica nanoparticles (PEI-MSNs), without any anti-cancer drug, very potently kill multiple GSC lines cultured in stem cell conditions. Very importantly, PEI-MSNs did not affect the survival of established GB cells, nor other types of cancer cells cultured in serum-containing medium, even at 25 times higher doses. PEI-MSNs did not induce any signs of apoptosis or autophagy. Instead, as a potential explanation for their lethality under stem cell culture conditions, we demonstrate that the internalized PEI-MSNs accumulated inside lysosomes, subsequently causing a rupture of the lysosomal membranes. We also demonstrate blood–brain-barrier (BBB) permeability of the PEI-MSNs in vitro and in vivo. Taking together the recent indications for the vulnerability of GSCs for lysosomal targeting and the lethality of the PEI-MSNs on GSCs cultured under stem cell culture conditions, the results enforce in vivo testing of the therapeutic impact of PEI-functionalized nanoparticles in faithful preclinical GB models.

## 1. Introduction

Glioblastoma (GB) is the most common, aggressive, and lethal form of primary brain tumors in adults [1,2]. The prognosis of patients affected by GB remains limited with a median survival of approximately 12–18 months [3]. The current clinical practices for patient treatments include surgery and chemo- and radiotherapy. The treatments are challenged by major complications because of the highly invasive nature of GB cells, intratumoral heterogeneity, and the intrinsic resistance of GB cells towards therapies [4]. It has been reported that even after surgery and chemo- and radiotherapy, GB cells invade neighboring normal brain, leading to currently incurable recurrence in patients [5]. Current therapeutic approaches leave the resistant and aggressive sub-population of GB cells, glioblastoma stem cells (GSCs), untreated [6]. GSCs have phenotypically distinctive characters: the ability to differentiate, self-renew, and form new tumors. GSCs are one of the main causes of resistance, recurrence, and mortality in GB; thus, novel therapeutic approaches are needed to target the GSC population [7].

During recent years, silica-based nanoparticles have gained vast attention in therapy, diagnosis, and theranostics [8]; albeit, limited examples exist specifically for brain cancer treatment. A few promising studies have been published to date [9,10,11,12,13,14], suggesting that their advantageous characteristics could be exploitable also in this therapeutic area. The most prominent advantage of the specific class of mesoporous silica nanoparticles (MSNs) is perhaps their drug delivery capability, being able to efficiently carry high payloads of especially poorly soluble drugs. Further, MSNs can flexibly be equipped with controlled release functions, ability to cross biological barriers, e.g., the cell membrane, and, in the best-case scenario, deliver the drug in a targeted fashion [15,16,17,18,19,20,21]. The silica surface is inherently negatively charged, which generally does not maximize attraction to the negatively charged cellular membranes. Thus, hyperbranched polyethyleneimine (PEI), a polycationic polymer with a high abundance of amino groups [22], is widely applied as a coating for enhancing the cellular uptake of nanoparticles to achieve efficient delivery of therapeutic payloads to cells [23,24,25]. Whereas free PEI is unselectively toxic to established cancer cells in culture, incorporation of PEI to the surface of nanoparticle carriers has been shown to eliminate its toxicity against thus far tested cancer cells [23]. In addition to enhancing cellular uptake, PEI is widely believed to promote endosomal escape via the proton-sponge effect [26,27]. The proton-sponge hypothesis suggests that cationically surface-functionalized nanoparticles cause endo/lysosomal swelling by intake of water molecules, eventually leading to the disintegration of the endo/lysosomal membranes [28,29].

In the present study, we set out to evaluate the in vitro and in vivo potential of PEI surface-functionalized MSNs (PEI-MSNs) (Figure 1) as drug carriers for GB therapy. To our surprise, PEI-MSNs without carrying any drugs induced death of GSCs grown under stem cell conditions, but not of established GB cells, or other tested cancer cell lines cultured in standard conditions. Confocal and transmission electron microscopy analysis of the PEI-MSN-treated GSCs indicated involvement of the “proton-sponge mechanism” [28,29], leading to rupture of the lysosomal membrane as a primary cell death mechanism. Additionally, to deduce the potential of this mechanism to be exploited in a therapeutic setting, we showed successful penetration of PEI-MSNs through BBB both in vitro and in vivo. Collectively, the discovery of the intrinsic role of PEI-MSNs in eradicating otherwise highly resistant GSCs presents a novel vulnerability to be exploited for brain cancer (GB) treatment.

## 2. Materials and Methods

Unless otherwise noted, all reagent-grade chemicals were used as received, and Millipore water was used in the preparation of all aqueous solutions. Cetylmethylammonium bromide (CTAB, AR) was purchased from Fluka (Buchs, St. Gallen, Switzerland). 1,3,5-Trimethyl-benzene (TMB, 99%) was purchased from ACROS (Fair Lawn, NJ, USA). Decane (99%) was purchased from Alfa Aesar (Ward Hill, MA, USA). Anhydrous toluene (AR), ethylene glycol (AR), tetraethyl orthosilicate (TEOS, AR), 3-aminopropyltriethoxysilane (APTES, AR), and NH_4_OH (30 wt%, AR) were purchased from Sigma Aldrich (Saint Louis, MO, USA). Aziridine was used in the preparation for hyperbranched surface modification of the MSNs and purchased from Menadiona S.L.Pol. Industrial company (Palafolls, Barcelona, Spain).

### 2.1. Preparation and Characterization of Hyperbranched PEI-Functionalized Mesoporous Silica Nanoparticles (PEI-MSNs)

MSNs were prepared according to a protocol from our previously published work [30]. The schematic presentation of their preparation is shown in Figure 1. The MSNs were prepared by co-condensation of TEOS and APTES as silica sources. Briefly, a mixed solution was prepared by dissolving and heating CTAB (0.45 g) in a mixture of DI water (150 mL) and ethylene glycol (30 mL) at 70 °C in a reflux-coupled round flask reactor. Ammonium hydroxide (30 wt%, 2.5 mL) was introduced to the reaction solution as the base catalyst before TEOS (1.5 mL) and APTES (0.3 mL) was added to initiate the reaction. Decane (2.1 mL) and TMB (0.51 mL) were used as swelling agents before the addition of the silica sources. Decane was added 30 min before TMB and after the addition of TBM, the synthesis solution was mixed for 1.5 h. The molar ratio of used reagents in the synthesis of the MSNs was 1TEOS:0.19APTES:0.18CTAB:0.55TMB:1.6 decane:5.9NH_3_:88.5 ethylene glycol:1249H_2_O. For inherent fluorophore labeling of the MSNs, TRITC was pre-reacted with APTES in a molar ratio of (APTES:TRITC) 3:1 in ethanol (0.5 mL) under vacuum for 2 h. Subsequently, the pre-reaction solution was added to the synthesis solution before the addition of TEOS. The reaction was allowed to proceed for 3 h at 70 °C. Then, the heating was stopped where after the as-synthesized colloidal suspension was aged at 70 °C without stirring for 24 h. After the suspension was cooled to room temperature, the suspension was separated by centrifugation. After collecting the particle precipitate, the template removal was carried out by the ion-exchange method. Briefly, the collected particles were extracted three times in ethanolic NH_4_NO_3_ solution, washed with ethanol [10], and resuspended in DMF for long-term storage. The surface modification of the MSNs with hyperbranched PEI by surface-initiated polymerization was carried out according to an in-house-established protocol [22]. To initiate PEI polymerization from the MSNs surfaces, aziridine was used as a monomer with toluene as solvent, in which the MSN substrate was suspended in the presence of catalytic amounts of acetic acid. The suspension was refluxed under atmospheric pressure overnight at RT, filtered, washed with toluene, and dried under vacuum at 313 K. Henceforth, the obtained nanoparticles are abbreviated as PEI-MSNs. Full redispersibility of the dried, extracted, and surface-functionalized MSNs was confirmed by redispersion of the dry particles in HEPES buffer at pH 7.2 and subsequent dynamic light scattering (DLS) measurements (Malvern ZetaSizer NanoZS). The fine architecture of the nanoparticles was further confirmed by transmission electron microscopy (Jeol JEM-1200EX electron microscope) operated at 80 kV. The success of the surface polymerization was confirmed by zeta potential measurements (Malvern ZetaSizer NanoZS).

### 2.2. Cell Culture

Established human GB cell lines U87MG (a gift from Ari Hinkkanen, University of Eastern Finland, Joensuu, Finland), A172 (University of Helsinki, Helsinki, Finland) were cultured in DMEM (Sigma-Aldrich, Saint Louis, MO, USA), and T98G (VTT Technical Research Centre, Turku, Finland in 2010) was cultured in Eagle MEM (Sigma-Aldrich, Saint Louis, MO, USA), supplemented with 10% heat-inactivated FBS (Biowest, Rue du Vieux Bourg, France), 2 mM L-glutamine, and penicillin (50 U/mL)/streptomycin (50 µg/mL). The patient-derived GSCs BT-3-CD133^+^, BT-12, and BT-13 [31] were previously analyzed by total RNA sequencing (data accessible on the Gene Expression Omnibus, submission reference GSE169418). According to the molecular classification of glioblastomas [32], these cells belong to the Mesenchymal subtype. Patient-derived GSCs were cultured in Dulbecco’s modified Eagle’s medium with Nutrient Mixture F-12 (DMEM/F12, Gibco, Thermo Fisher Scientific, Waltham, MA, USA) supplemented with 2 mM L-glutamine, 2% B27-supplement (Gibco, Thermo Fisher Scientific, Waltham, MA, USA), 50 U/mL penicillin and 50 µg/mL streptomycin, 0.01 µg/mL recombinant human fibroblast growth factor-basic (FGF-b, Peprotech, Cranbury, NJ, USA), 0.02 µg/mL recombinant human epidermal growth factor (EGF, Peprotech, Cranbury, NJ, USA), and 15 mM HEPES-buffer. The blood–brain tumor barriers (BBTB) were established as previously described [26]. Immortalized mouse endothelial cells from brain microvessels (bEND3) were maintained in DMEM (Lonza, Basel, Switzerland) supplemented with 10% decomplemented FBS (Lonza, Basel, Switzerland), 2 mM L-glutamine (Sigma-Aldrich, Saint Louis, MO, USA), and penicillin:streptomycin (50 U/mL and 50 µg/mL respectively). Mouse immortalized astrocytes (HIFko) were maintained in Basal Eagle Medium 1 (BME-1, Sigma-Aldrich, Saint Louis, MO, USA) supplemented with 5% decomplemented FBS (Lonza, Basel, Switzerland), 1 M HEPES (Sigma-Aldrich, Saint Louis, MO, USA), 2 mM L-glutamine (Sigma-Aldrich, Saint Louis, MO, USA), 100 mM sodium pyruvate (Sigma-Aldrich, Saint Louis, MO, USA), 3 g D-glucose, and penicillin:streptomycin (50 U/mL and 50 µg/mL respectively). MDA-MB-231 and HeLa cells (a gift from Turku Bioscience Center, University of Turku and Åbo Akademi University, Turku, Finland) were cultured in Dulbecco’s modified Eagle’s medium (DMEM) (Sigma-Aldrich, Saint Louis, MO, USA) supplemented with 10% fetal bovine serum, 2mM L-glutamine, and 1% penicillin-streptomycin (*v/v*). All cell lines were kept in a humidified atmosphere of 5% CO_2_ at 37 °C. For colony growth and microscopy, the GSC populations were cultured as monolayers on Matrigel-coated (Becton Dickinson, Franklin Lakes, NJ, USA) dishes.

### 2.3. Western Blotting and Antibodies

BT-12 cells were treated with 10 µg/mL PEI-MSNs for 24 h and 48 h. They were lysed in 2× Laemmli buffer (4% SDS, 20% glycerol, 120 mM Tris) and resolved by SDS-PAGE gel (BioRad, Hercules, CA, USA). Proteins were transferred to nitrocellulose membranes (BioRad, Hercules, CA, USA). Membranes were blocked with 5% milk-TBS and incubated with a required dilution of primary and 1:5000 dilution of secondary antibodies in 5% Milk-TBS-Tween 20 for the required duration of time and visualized with Odyssey (LI-COR Biosciences, Lincoln, NE, USA). The following antibodies were used: PARP-1 (sc-7150, 1:1000) and P62 (sc-28359, 1:500 dilution) from Santa Cruz Biotechnology (Dallas, TX, USA), cPARP (ab32064) (1:1000 dilution) from Abcam (Cambridge, UK), and LC3-β (2775 s) (1:1000) from Cell Signaling (Danvers, MA, USA). Antibody against β-actin (sc-47778) (1:10,000 dilution), the loading control, was from Santa-Cruz Biotechnology (Dallas, TX, USA). Secondary antibodies were purchased from LI-COR (Lincoln, NE, USA), mouse (926-32212) and rabbit (926-68021). For stripping of membranes, Re-blot Plus (Millipore, Burlington, MA, USA) solution was used.

### 2.4. Flow Cytometry

To further study the apoptosis of BT-3-CD133^+^ cells, an optimized number of cells (2 × 10^5^ cells) were seeded into 6-well plates (Sigma-Aldrich, Saint Louis, MO, USA), and the cells were treated with PEI-MSNs for 24 h. Cells were then harvested, washed twice with PBS, and once with 1 × Binding Buffer ((5 mM HEPES, 70 mM NaCl, 2.5 mM CaCl_2_, pH 7.4) in 2%FCS/PBS (*w*/*v*), 0.01% NaN_3_). Cells were stained with Annexin V-FITC (BD Pharmingen, San Jose, CA, USA) and incubated at RT for 20 min. Cells were then washed twice with Binding buffer. Twenty seconds prior to analysis with the FACS Calibur system, propidium iodide (PI; BD Pharmingen) was added to the sample. The data were analyzed with CellQuest Pro (BD Biosciences) or FlowJo (TreeStar Inc., Ashland, OR, USA).

### 2.5. Colony Formation Assay

An optimized number of cells (3 × 10^3^ to 10 × 10^3^) were seeded in 24-well plates (Sigma-Aldrich, Saint Louis, MO, USA) and allowed to attach. After 24 h, the cells were treated with 1–50 µg/mL of PEI-MSNs. After 72 h, the medium was replaced with fresh medium and the cells were incubated for another 72 h or until the control well was confluent. Cell colonies were fixed with methanol dilutions and stained with 0.2% crystal violet (CV) solution in 10% ethanol for 15 min at room temperature. Plates were dried and scanned with Epson Perfection V700 Photo scanner (Suwa, Nagano prefecture, Japan). Quantifications were performed with ImageJ by using the Colony area plugin [33]. Data were normalized and presented as a percent of the control.

### 2.6. Light Microscopy

#### 2.6.1. Immunofluorescence (Early Endosomes and Lysosomes)

BT-12 GSCs were grown as monolayers on Matrigel-coated glass coverslips. BT-12 GSCs were treated with 10 µg/mL of PEI-MSNs conjugated with TRITC (Tetramethylrhodamine-isothiocyanate) for 48 h. BT-12 GSCs were fixed with 4% PFA (paraformaldehyde) for 10 min. The cells were permeabilized using 0.1% Triton X-100 for 10 min and blocked with horse serum. The 1° anti-EEA1 (goat) antibody for recognition of early endosomes (Santa Cruz Biotechnology, Dallas, TX, USA) was prepared (1:100) in PBS (10% horse serum). The 1° anti-LAMP-1 (mouse) antibody for recognition of lysosomes (Abcam, Cambridge, UK) was prepared (1:100) in PBS (10% horse serum). Antibody incubation was performed overnight at +4 °C. The cells were washed three times with PBS; Alexa 488 secondary (Anti-goat and anti-mouse) antibody (Sigma-Aldrich, Saint Louis, MO, US) in PBS was added to the cells at RT for 1 h. The cells were mounted on coverslips using VECTASHIELD (4′,6-diamidino-2-phenylindole). The microscopy setup consisted of a Zeiss 780 (Zeiss, Jena, Germany) confocal microscope, PMT, and 100× oil objective. DAPI was excited by 405 lasers and emission was collected in the blue channel. Alexa 488 (early endosomes and lysosomes) was excited with a 488 nm argon laser and emission was collected by the green channel (510–550 nm). The TRITC-labeled PEI-MSNs were excited by a 561 nm laser and the emissions were collected (575–610 nm).

#### 2.6.2. Mitochondrial Staining

BT-12 GSCs were grown as monolayers on Matrigel-coated glass coverslips and further treated with 10 µg/mL of PEI-MSNs conjugated to FITC (fluorescein isothiocyanate) for 48 h. Cell medium (0.5 mL) was collected, mixed with 0.2 μL of Mitotracker Orange^®^ (Thermo Fisher Scientific Inc., Waltham, MA, USA), and returned to the cells drop-by-drop. The cells were finally incubated for 20 min at 37 °C. The cells were washed 3 times with PBS, fixed for 10 min with 4% PFA, and mounted using VECTASHIELD (4′,6-diamidino-2-phenylindole) on glass slides for microscopy. The microscopy setup consisted of a Zeiss 780 (Zeiss, Jena, Germany) confocal microscope, PMT, and 100× oil objective. DAPI was excited by 405 lasers and emissions were collected in the blue channel. FITC-conjugated PEI-MSNs were excited with a 488 nm argon laser and emissions were collected by the green channel (510–550 nm). The Mitotracker Orange^®^ was excited by a 561 nm laser and emissions were collected at 575–610 nm.

### 2.7. Transmission Electron Microscopy (TEM)

BT-12 GSCs were grown as monolayers on Matrigel-coated glass coverslips and further treated with 10 µg/mL of PEI-MSNs 24 and 72 h. The BT-12 GSCs were fixed with 5% glutaraldehyde s-collidine buffer, post-fixed with 2% OsO_4_ containing 3% potassium ferrocyanide, dehydrated with ethanol, and flat embedded in a 45359 Fluka Epoxy Embedding Medium kit (Fluka, Buchs, St. Gallen, Switzerland). Thin sections were cut using an ultramicrotome to a thickness of 100 nm. The sections were stained using uranyl acetate and lead citrate to enable detection with TEM. The sections were examined using a JEOL JEM-1400 Plus transmission electron microscope operated at an 80 kV acceleration voltage [34].

### 2.8. In Vitro Blood–Brain Tumor Barrier

Murine blood–brain tumor barriers (BBTB) in a dish were established according to a previously published protocol [26]. Briefly, immortalized mouse brain microvessel endothelial cells (bEND3) were co-cultured in Transwell inserts with immortalized mouse astrocytes (HIFko) in serum-free conditions. After the 6 days required for the formation of a tight artificial BBB, the inserts were transferred to 6-well plates previously seeded with BT-12 GSCs on glass coverslips, in stem cell medium, as the final step required to form the BBTB. To evaluate the nanoparticles transportation through the BBTB and the targeting of BT12 GSCs, 100 ng of PEI-MSI were added on the endothelial side. After 24 h, each cell type forming the BBTB were stained by adding LysoTracker Red DND-99 to the media according to the manufacturer’s recommendations (Invitrogen, Carlsbad, CA, USA) before fixation with ice-cold 4% PFA (10 min) and nuclear counterstaining with DAPI (1 µg/mL, Sigma). BT-12 coverslips and Transwell membranes containing both bEND3 and HIFko cells were eventually cut and mounted on Mowiol 4-88 (Sigma-Aldrich, Saint Louis, MO, USA) and imaged on a Zeiss LMS880 confocal microscope.

To quantify the cell viability of the bEND3, astrocytes, and BT-12 GSCs from the BBTB after 72 h, cells were gently detached with accutase (Sigma-Aldrich, Saint Louis, MO, USA) collected, counted, and 5 × 10^5^ cells/mL were transferred in a 96-well plate. A total of 10 μL of 3-(4,5-Dimethylthiazol-2-yl)-2,5-Diphenyltetrazolium Bromide (MTT; 5 mg/mL in PBS) was added to the cells before incubating for 2 h at 37 °C. Eventually, the cells were lysed (10% SDS, 10 mM HCl o/n), and the absorbance was measured at 540 nm using Multiskan Ascent software version 2.6 (Thermo Fisher Scientific, Waltham, MA, USA). Results were expressed as the % of absorbance relative to the control (untreated BBTB cells).

### 2.9. In Vivo Procedures

All experiments involving animals were authorized by the National Animal Experimental Board in Finland (Helsinki, Finland), under the licenses ESAVI/6285/04.10.07/2014 and ESAVI/403/2019.

Intracranial implantation of U87MG-GFP (*n* = 10) or BT-12 (*n* = 10) cells was performed as previously described [15]. Briefly, 8-week-old female NMRI:Rj nude mice were implanted with 10^5^ cells in 10 µL in the right striatum. After 20 days of tumor growth, 100 µg of PEI-MSNs in PBS were injected in the caudal vein (100 µL, *n* = 5) or intranasally (3 dosages of 5 µL given every two hours, *n* = 5). After 8 h, animals were euthanized, and brains were snap-frozen in −50 °C isopentane (Honeywell, Charlotte, NC, USA). Brain cryosections (9 µm) were cut using a cryotome (Thermo Fisher Scientific, Waltham, MA, USA), collected on Superfrost Ultra slides (Thermo Fisher Scientific, Waltham, MA, USA), and fixed in an ice-cold 4% PFA bath. Brain microvessels were stained overnight using a rat anti-mouse PECAM-1/CD31 (1:400, 553370, BD Pharmingen, Franklin Lakes, NJ, USA). Cell nuclei were counterstained with DAPI (1 µg/mL, Sigma), where after the samples were mounted with Mowiol 4-88 and imaged on a Zeiss LMS880 confocal microscope. The in vivo protocol was conducted twice.

## 3. Results

Examination of the hydrodynamic size and ζ-potential values of the PEI-functionalized MSNs (Figure 1) in HEPES buffer solution (25 mM, pH 7.2) at a concentration of 0.1 mg/mL yielded a hydrodynamic mean size of 124 ± 12 nm with a low polydispersity index (PDI) value of 0.09, indicating a monodispersed colloidal suspension of PEI-MSNs. In addition, the ζ-potential value of the PEI-MSNs in HEPES buffer (+39 ± 4 mV) ascertained the high net positive charge on the MSN surfaces owing to successful surface modification with PEI. The mesoscopic ordering of the MSN structure before the surface modification was examined by TEM imaging of the samples (Appendix A). As presented in the TEM micrographs, spherical particles with an approximate size of 50 nm with a porous structure were obtained.

### 3.1. PEI-MSNs Exhibit Specific Toxicity towards GSCs Cultured under Stem Cell Conditions

The PEI-MSNs (Figure 1) were applied (1–50 µg/mL) to T98G (established GB cell line), BT-3-CD133^+^, BT-12, and BT-13 (patient-derived GSCs) cells (Appendix A) and the colony formation was followed by crystal violet staining (Figure 2A). The efficiency of colony formation was quantified by using the “ColonyArea” ImageJ plugin [33] (Figure 2B). The exposure of PEI-MSNs to GSC cells cultured under stem cell conditions resulted in pronounced inhibition of colony growth even at particle concentrations as low as 1 µg/mL.

In contrast, the GB cell lines T98G, U87MG, and A172, or MDA-MB-231 breast carcinoma and HeLa cervical carcinoma cells cultured under standard conditions, showed reduced colony growth in comparison to the control-treated cells only at a very high PEI-MSN concentration (50 µg/mL) (Figure 2B, Figure 3 and Appendix A). These observations can be correlated to the well-reported fact that PEI can induce non-specific toxicity to cells if applied at very high concentrations [35,36,37,38,39].

Importantly, we further verified that MSNs without PEI did not cause cytotoxicity at the concentration range of 1–50 µg/mL (Figure 4 and Appendix A). Thus, our results reveal that PEI functionalization results in induction of death of patient-derived GSCs cultured under stem cell conditions.

### 3.2. GSCs Show No Induction of Apoptosis or Autophagy after PEI-MSN Treatment

We further investigated the role of PEI-MSNs in the induction of cell death of patient-derived GSCs cultured under stem cell conditions. The BT-12 and BT-13 cells were the most sensitive to PEI-MSN-elicited growth inhibition (Figure 2B). Based on these results, we selected the BT-12 GSCs for an in-depth analysis. Initially, the possible roles of autophagy or apoptosis in cell death were investigated by studying the cleavage of PARP-1 as an apoptosis marker [40,41,42]. BT-12 GSCs showed detectable PARP cleavage already in control conditions and PEI-MSN treatment for 24 h or 48 h did not increase the expression levels of this apoptotic biomarker (Figure 5A). Similar results were obtained with flow cytometry after PI/Annexin staining (Appendix A).

To determine the possible effect on autophagy, we studied the expression of specific autophagy biomarkers P62 and LC3B [43,44,45,46] in PEI-MSN-treated cells. The biomarker expression of the PEI-MSN-treated cells was similar to that of control cells, and no significant increase in autophagy-related biomarkers was observed (Figure 5B). Thereby, these results suggest that cell death was not mediated either by apoptosis or autophagy.

### 3.3. PEI-MSNs Localize within the Cytoplasmic Space and Lysosomes

To understand the potential cell death mechanism in BT-12 GSCs, we studied the intracellular localization of PEI-MSNs by confocal microscopy. We selected early endosomes (EEA1), nucleus (DAPI), mitochondria (Mitotracker), and lysosomes (LAMP-1) to comprehend the PEI-MSNs’ interactions with the intracellular organelles. Figure 6 shows that upon 48 h treatment, the PEI-MSNs mostly co-localized with the lysosomal marker (LAMP-1) in the BT-12 GSCs.

PEI-MSNs were not localized within the nucleus (Figure 6 and Appendix A). Besides, a non-significant number of PEI-MSNs overlapped with either early endosomes (EEA1) or mitochondria. Individual PEI-MSNs (50 nm in diameter) were beyond the limit of resolution by confocal microscopy [47]. Despite that, the co-localization of the PEI-MSNs with the lysosomal marker (LAMP-1) was very frequently observed (Figure 6 and Appendix A). This was anticipated, given that nanoparticles typically enter cells by endocytosis after which they are transported to lysosomes [34,48,49].

### 3.4. PEI-MSNs Cause Lysosomal Membrane Rupture in GSCs, Leading to Cell Death

Detection of the endosomal escape of PEI-MSNs is beyond the detection limit of confocal microscopy. Therefore, to detect a possible “proton-sponge effect” via membrane destabilization, we performed subsequent transmission electron microscopy (TEM) imaging of PEI-MSN-treated BT-12 cells. TEM imaging of the treated cells revealed the widespread dissemination of PEI-MSNs throughout the cytoplasm (Figure 7A–D). Moreover, we observed three distinct types of lysosomal accumulation of PEI-MSNs in the BT-12 GSCs: (1) lysosomes filled with PEI-MSNs (Figure 7A); (2) empty lysosomes with PEI-MSNs localized in the proximity to a lysosomal membrane (Figure 7B); and (3) lysosomes semi-filled with PEI-MSNs (Figure 7C,D). In summary, PEI-MSNs appeared to accumulate mostly in lysosomes whereas individual PEI-MSN particles were also observed in non-lysosomal spaces (Figure 7). Most relevant to the proposed cell killing mechanism by PEI-MSNs, we detected permeabilization of the lysosomes and the potential escape of PEI-MSNs from the damaged lysosomes by TEM (yellow arrows in Figure 7A–D). Similar lysosomal escape of PEI-MSNs was not observed in established MDA-MB-231 cells, as the lysosomal structures remained confined and there was no visible damage to the cellular structures (Appendix A).

### 3.5. PEI-MSNs Cause Morphological Abnormalities in GSCs

With TEM, we also observed morphological abnormalities (Figure 8A,B) and structural damage of mitochondria (Figure 8C,D) in the PEI-MSN-treated BT-12 GSCs cultured under stem cell conditions. On the other hand, we did not observe PEI-MSNs permeating the nuclear space (Appendix A). The easily recognizable abnormalities in the ultrastructure as compared to the non-treated cells included loss of lysosomal integrity, mitochondrial swelling, and rupture of cristae (Figure 8A–H). The process of endosomal trafficking begins with early endosomes and the endosomal payload can be either recycled to the plasma membrane via recycling endosomes, or it can advance to the late endosomes and lysosomes for degradation [50,51,52]. The proton sponge hypothesis indicates that PEI functionalization promotes escape from the endolysosomal pathway through rupture of the membrane. Numerous studies propose that membrane permeabilization occurs in the lysosomes [53,54]. Lysosomal membrane destabilization can lead to a triggered discharge of lysosomal enzymes to the cytoplasm, which eventually can cause cell death [55,56]. In the light of this knowledge, our evidence from confocal and TEM imaging suggests that the lysosomal membrane disruption by the PEI-MSNs could be a potential mechanism leading to cell death in BT-12 GSCs.

### 3.6. PEI-MSNs cross the Neurovascular Unit In Vitro and In Vivo

To validate whether PEI-MSNs could in principle be used in future for targeting GB in vivo, we screened their permeability through an in vitro model of BBTB [57]. This model establishes a mimic of the BBB in a brain tumor context by first co-culturing mouse brain microvascular endothelial cells and astrocytes, growing in serum-free conditions and pseudocontact through the 3 µm pore of the membrane of the Transwell inserts (Appendix A). Once the endothelial cells formed a tight monolayer, inserts were placed on BT-12 GSCs to finalize the BBTB and 100 ng of PEI-MSNs were added on the *blood* side of the inserts (Appendix A). The passage of the PEI-MSNs through the BBTB was followed by confocal microscopy, with lysotracker-red fluorescent dye used to label the lysosomes (Appendix A). After 24 h, PEI-MSNs were still detected on the endothelial cells and astrocytes and co-localized with lysosomes (Figure 9A,B). Interestingly, PEI-MSNs were also abundantly detected on the other side of the BBTB, at the surface and within BT12 gliospheres (Appendix A). In particular, PEI-MSNs were colocalized with lysotracker-red positive lysosomes in the BT-12 GSCs (Figure 9C). After 3 days, endothelial cells, astrocytes, and BT-12 cells were removed from the Transwells and their viability was measured by MTT. BT-12 gliosphere viability was 31% lower compared to untreated BT-12 cells isolated from the control BBTB (Figure 9D). The viability of endothelial cells and astrocytes was not significantly affected compared to the untreated cells (−3% and −6%, respectively), suggesting selective toxicity towards the BT-12 GSCs under these culture conditions.

We then proceeded with the in vivo evaluation of the PEI-MSN passage through the neurovascular unit and brain distribution in immunocompromised mice implanted with BT-12 or U87MG-GFP cells. Due to the relatively small diameter (50 nm) of the PEI-MSNs, we evaluated two different administration routes, i.e., the classical caudal vein route requiring BBB permeability [58] or an intranasal dosage [58], which is based on the direct accessibility to the central nervous system due to fenestrated nasal epithelial tissue and endothelial blood vessels localized before the cribriform plate bone, at the junction of the nasal cavity and the frontal bone [59]. This discontinuity in the cranial bone in addition to the bypass of the highly selective BBB grant access to the brain parenchyma through the olfactory neuron endings. Therefore, compounds or molecules are typically endocytosed by the cilia, can travel inside the axons, and reach the central nervous system starting from the olfactory bulb. Intranasal administration of PEI-MSNs (35 µg in 5 µL of PBS) was distributed drop by drop, alternating between the nostrils. The procedure was repeated 3 times every two hours. Animals were euthanized 2 h after completing the intranasal dosage and eight hours after IV injections of the PEI-MSNs, and brains were collected. We then verified the distribution of the particles in different areas of the brain parenchyma and the GB xenografts. We observed PEI-MSNs associated with or outside the brain blood capillaries in different regions of the cerebral cortex following IV injections (Figure 10A–C). For the intranasally administered mice, no preferential distribution of the PEI-MSNs was observed around blood vessels, but nanoparticles could still be detected within various regions of the brain parenchyma, including the posterior parts of the encephalon, such as the hippocampus (Figure 10B,C). We verified the distribution of the PEI-MSNs in the olfactory bulb of both the IV and intranasally treated animals. We could observe a very high density of nanoparticles in the olfactory bulb of the intranasally administered mice (Figure 10D), supporting the brain accessibility of the PEI-MSNs through the vomeronasal nerves. As PEI-MSNs can also cross the BBB when delivered intravenously, we could detect PEI-MSNs in the olfactory bulb tissue of the IV-injected mice at densities similar to what was observed in the rest of the brain (Figure 10D). We eventually verified the presence of nanoparticles in the BT-12 and U87MG-GFP intracranial tumors (Figure 10E,F). Interestingly, PEI-MSNs could be observed in tumors after both IV and intranasal administration, although the IV-injected animals seemed to exhibit a better intratumoral distribution with more nanoparticles observed within the tumor tissue compared to the intranasal delivery (Figure 10E,F).

## 4. Discussion

The starting point of this study was to develop a PEI-functionalized MSNs nanocarrier for CNS delivery of siRNA (RNAi therapy) [24] or small-molecule drugs for the treatment of GB tumors. The PEI-MSNs were designed to overcome challenges with BBB permeability [10], consequently increasing the accumulation of the therapeutic compounds at the tumor site [60]. As a flexible functionalization platform for MSN-based delivery systems, we used a PEI surface polymer coating, promoting cellular uptake by providing an overall positive surface charge, as well as increasing the reaction sites for potential further conjugation of active moieties for BBB permeability and GB targetability. While evaluating our different siRNA/drug delivery system designs on the GB cells, including patient-derived GSCs, we discovered high lethality in patient-derived GSC cell lines cultured under stem cell conditions when treated with the PEI-MSN controls (without drug). Notably, MSNs without a PEI coating did not exhibit this effect.

The well-known property of PEI in enhancing cellular uptake due to the high positive charge density is frequently exploited in gene transfection, whereby PEI is complexed with nucleic acids to form a nanoparticle-like assembly. On the other hand, free PEI is highly water-soluble, and thus if dissolved into cell media, by itself causes non-specific cytotoxicity via induction of membrane damage and apoptosis [35,37,38,39]. Nevertheless, based on our data, when PEI is used as a nanoparticle coating, the unselective cytotoxicity of free PEI [24] can be circumvented, or at least significantly repressed. Indeed, already at exceptionally low doses (1 µg/mL), the PEI surface-functionalized MSNs, without any anti-cancer therapeutic, very potently inhibited cell growth of multiple GSC lines cultured under stem cell conditions—but not GB cells cultured under standard conditions. However, a limitation of our results arises from these different cell culture methods used for the established GB cell lines and GSCs. This could in theory affect the vulnerability of the GSCs to PEI-MSNs. On the other hand, we did not detect any toxic effects of PEI-MSNs on endothelial cells or astrocytes in the in vitro BBB model, whereas the PEI-MSNs significantly reduced the viability of GSCs, supporting their killing activity towards GSCs, even under similar experimental conditions. The other limitations of this study were that the results were reported for only three different patient-derived GSCs. Thus, it would be important to investigate if vulnerability of GSCs to PEI-MSNs can be validated throughout different stem cell populations within the tumor microenvironment. Regardless of these limitations, identification of strategies that kill GB cell populations may have relevance for designing future GB therapies.

From a mechanistic point of view, the results show that PEI-MSNs accumulate in the lysosomes of BT-12 cells after cellular uptake; which is a typical observation as MSNs often enter cells by endocytosis. Furthermore, we found that PEI-MSNs disrupt the lysosomal membrane. We hypothesized that this phenomenon occurred due to the so-called “proton-sponge effect” as PEI is believed to promote lysosomal escape via this mechanism [29,61,62]. This, in turn, leads to lysosomal membrane permeabilization, causing widespread localization of PEI-MSNs and lysosomal enzymes, such as cathepsins and other hydrolases, into the cytoplasmic space [55,56]. The release of lysosomal enzymes leads to cell death. Although it is not yet clear why GSCs would be particularly sensitive to lysosomal targeting, similar vulnerability in the GSCs was recently found to be induced by antihistaminergic drugs [31] or by targeting of MALT1 [63]. We did not find any indications of apoptotic cell death in the GSCs after PEI-MSN treatment, which suggests that following rupture, the cells may have died acutely by necrosis. Therefore, in addition to discovery of PEI-MSN-mediated GB cell killing, our data generally strengthens the theory of the lysosomal vulnerability of GSCs.

To test whether PEI-MSNs could be developed towards clinical treatment of GB, we demonstrated that PEI-MSNs can be successfully delivered across the BBB in vitro and in vivo. This is consistent with our earlier data indicating that PEI-MSNs can cross the BBB [10]. Administration in vivo via intranasal and intravenous routes exhibited widespread distribution of PEI-MSNs throughout the brain. Intranasal delivery showed heterogenous distribution of the PEI-MSNs in the brain, i.e., an extremely high concentration at the entrance points in the olfactory bulb and only a few PEI-MSNs in distant structures, such as the hippocampus and the tumor. However, this provides a proof-of-principle that the central nervous system can be reached through this non-invasive method; also, as the travel of the nanoparticles endocytosed at the nerve endings is certainly slower than the blood flow, longer timepoints could show an enhanced PEI-MSN distribution in the brain tissue. Intravenously injected PEI-MSNs were observed to be widespread in distant brain structures; hence, we propose intravenous injection as the superior dosing regimen, to be tested in the future in in vivo studies.

Drug-based nanomedicine approaches have been recently exploited to kill GB cells [64,65,66,67]. Apart from those studies, our results suggest that PEI modification imparts a new, intrinsic property to nanoparticles in killing of GSCs without any additional anti-cancer drug treatment when cultured specifically under stem cell conditions. We hypothesize that this phenomenon occurs through lysosome-associated pathways, ultimately leading to lysosomal membrane permeabilization through the proton-sponge effect. Taken together, the results reveal novel vulnerabilities of GSCs that could potentially be exploited for novel types of treatment regimens.

## 5. Conclusions

This study confirmed the inherent lysosomal vulnerability of glioblastoma stem cells cultured under stem cell conditions as therapeutic targets for PEI-functionalized nanoparticles. This discovery was conveyed by the intrinsic property of these nanoparticles to selectively kill patient-derived GSCs without any loaded drug. PEI-modified nanoparticles were shown to first accumulate inside lysosomes, subsequently causing a rupture of the lysosomal membrane specifically in GSCs. We hypothesize that this phenomenon occurs through the proton sponge effect due to the cationic nature of PEI-MSNs. This notion was further supported by the same phenomenon not being observed without a PEI coating on the nanoparticles, nor in GB cells without GSC characteristics. This suggests that even drug-resistant cells can be potently eradicated without drug therapy, which otherwise may contribute to the overall drug resistance. Given that the mode of eradication of GSCs via lysosomal disruption with drugs was only recently reported, this should be considered a noteworthy discovery in the intrinsic potency of nanoparticles in medicine. As a translational aspect, we further determined that the PEI-MSNs effectively cross the BBB in vitro and in vivo. In addition, this study implies compelling evidence for the therapeutic application of the proton-sponge effect by cationically surface modified nanoparticles to target cells with vulnerable lysosomes.

## Figures and Tables

**Figure 1 cancers-13-02631-f001:**
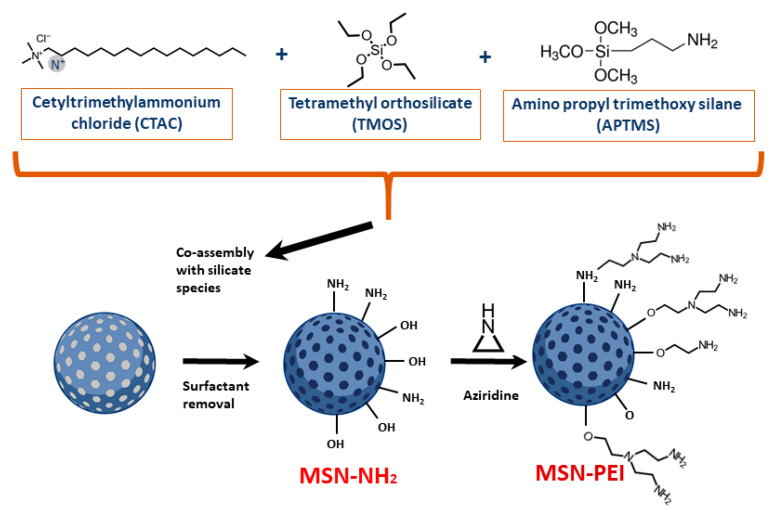
A graphic schematic presentation of a PEI-MSN.

**Figure 2 cancers-13-02631-f002:**
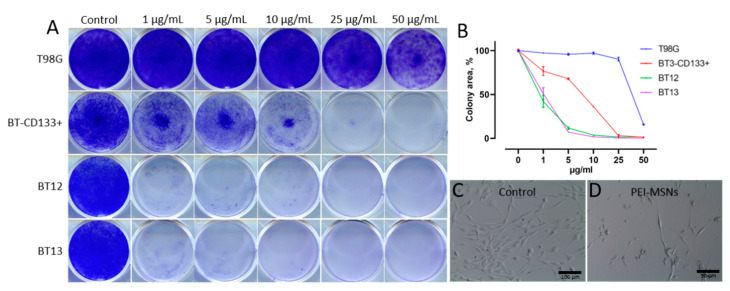
Death of patient-derived GSCs induced by PEI-MSNs. (**A**) Colony growth assay of T98G (GB cell line, cultured in standard conditions), BT-3-CD133^+^, BT-12, and BT-13 (GSCs, cultured in stem cell conditions) cells treated with 1–50 µg/mL of PEI-MSNs. (**B**) Quantification of colony growth by using the “ColonyArea”. (**C**,**D**) Representative images of BT-3-CD133^+^ (cultured in stem cell conditions) cells without (**C**) or with (**D**) 10 µg/mL PEI-MSN treatment.

**Figure 3 cancers-13-02631-f003:**
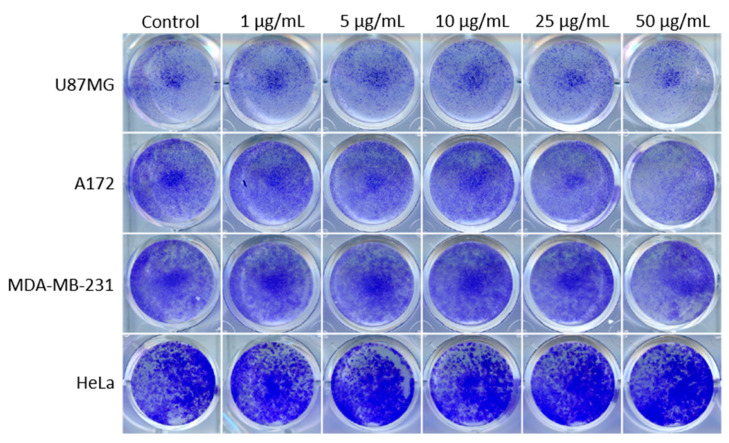
Colony growth of U87MG (human GB cells), A172 (human GB cells), MDA-MB-231 (human breast cancer cells), and HeLa cells (human cervical cancer cells) treated with 1–50 µg/mL of PEI-MSNs. All cells were cultured in standard cell culture conditions.

**Figure 4 cancers-13-02631-f004:**
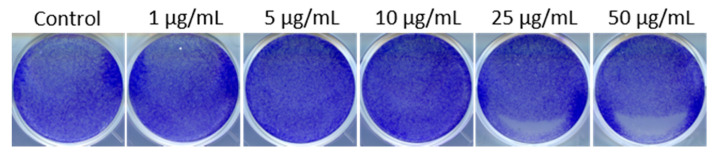
Colony growth assay of BT-12 (cultured in stem cell conditions) cells treated with 1–50 µg/mL of plain MSNs. MSNs without PEI functionalization do not show any cytotoxicity.

**Figure 5 cancers-13-02631-f005:**
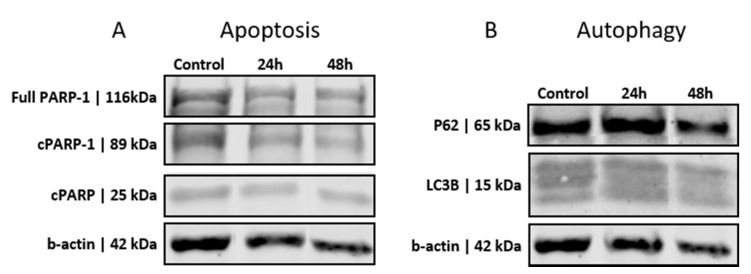
Western blot analysis of the apoptosis and autophagy biomarkers in BT-12 (cultured in stem cell conditions) GSCs treated with 10 µg/mL PEI-MSNs for 24 h and 48 h. (**A**) Expression levels of apoptotic biomarkers of the full-length PARP-1 and cPARP (89 kDA and 25 kDA). (**B**) Expression levels autophagy biomarkers P62 and LC3B (The original western blot figures are shown in Appendix A).

**Figure 6 cancers-13-02631-f006:**
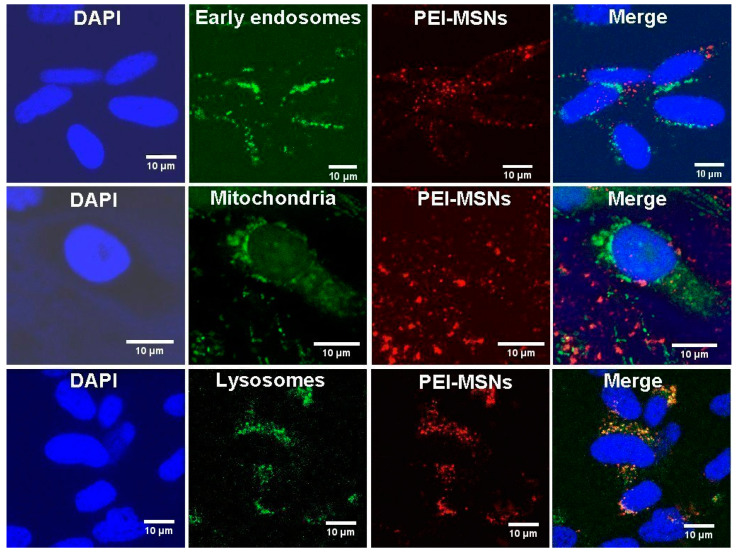
Localization of the PEI-MSNs in the treated BT-12 (cultured in stem cell conditions) GSCs by confocal microscopy. The intracellular localization of PEI-MSNs (red) was studied using markers of early endosomes, mitochondria, and lysosomes (green color). The nuclei were visualized by using DAPI (blue). The co-localization of the PEI-MSNs with lysosomes is seen in yellow.

**Figure 7 cancers-13-02631-f007:**
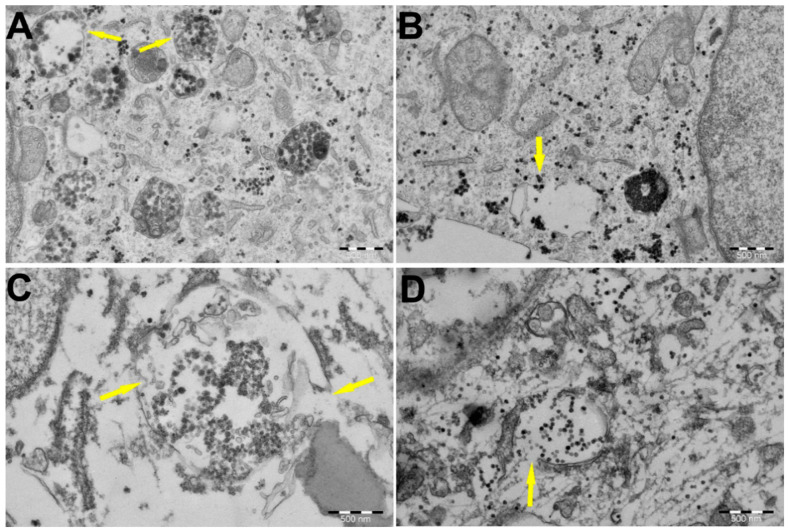
TEM imaging of the BT-12 (cultured in stem cell conditions) GSCs revealed the subcellular localization of the PEI-MSNs. PEI-MSNs were localized throughout the cells after 24 h. (**A**) Yellow arrows mark the potential endosomal escape of the PEI-MSNs via lysosomal membrane rupture. (**B**) PEI-MSNs in close proximity of a lysosome. (**C**,**D**) A close-up view of the affected lysosomes, suggesting a rupture of membranes and widespread localization of the PEI-MSNs.

**Figure 8 cancers-13-02631-f008:**
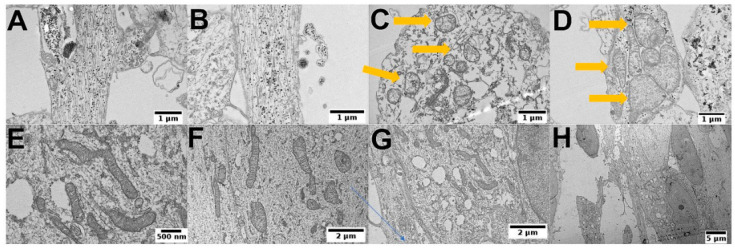
The cellular fate of PEI-MSN in the treated BT-12 (cultured in stem cell conditions) GSCs after 72 h. (**A**,**B**) Endosomally escaped PEI-MSNs can be observed widespread within the cells, tandem with loss of structural integrity of the cells. (**C**,**D**) Abnormalities in the mitochondrial morphology, mitochondrial swelling, and rupture of the cristae (arrows). TEM images of control BT-12 (cultured in stem cell conditions) GSCs without PEI-MSNs treatment. (**E**,**F**) A close overview of the intact mitochondrial structure. (**G**) Empty lysosomes and normal ultrastructure of BT-12 GSCs (cultured in stem cell conditions). (**H**) Overview of a control cell.

**Figure 9 cancers-13-02631-f009:**
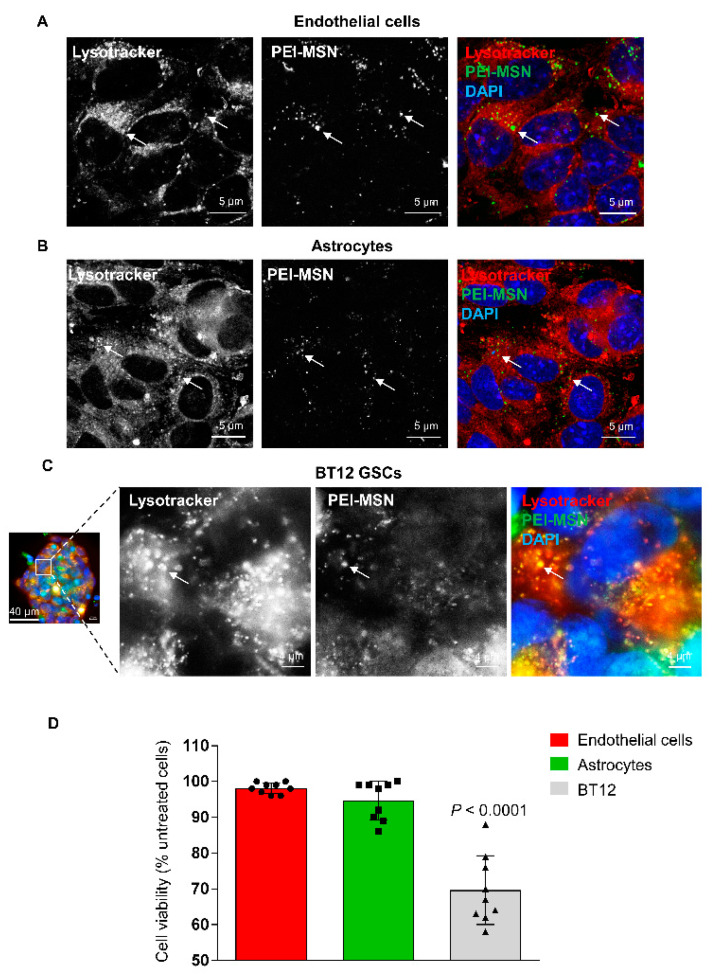
The PEI-MSNs cross the BBTB in vitro. (**A**–**C**) Confocal images of the colocalization of PEI-MSN (green) with the Lysotracker dye (red) in the endothelial (**A**), astrocytic (**B**), and GSC (**C**) compartments 24 h after addition of the PEI-MSN. Images in (**C**) are magnified from a BT-12 gliosphere (squared region of interest on the left panel). (**D**) Cell viability measured by MTT after 72 h. Values are normalized to the control conditions (cells isolated from untreated BBTB). *n* = 3, three pooled experiments. *p*-value calculated with one-way ANOVA with Sidak’s multiple comparison test.

**Figure 10 cancers-13-02631-f010:**
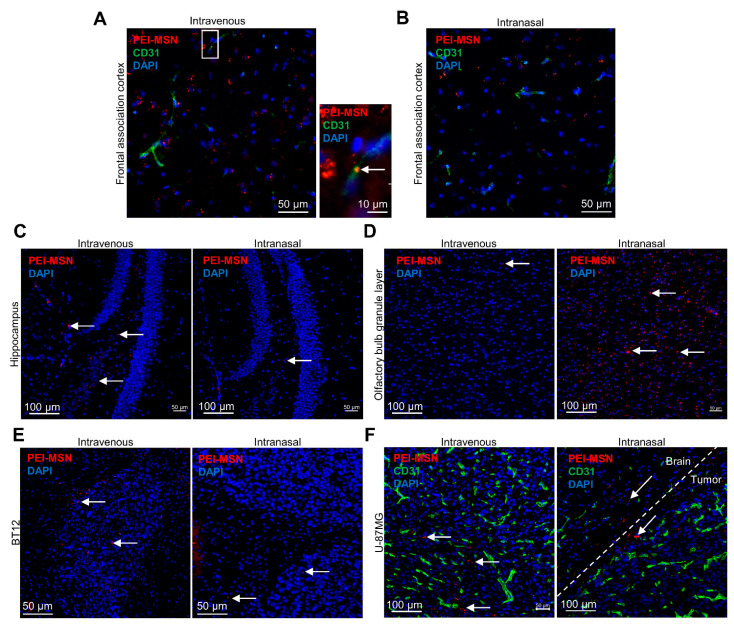
The PEI-MSNs can be delivered to the brain in vivo. (**A**,**B**) Confocal images of the colocalization of PEI-MSNs (red) with the endothelial cell marker CD31 (green) in the frontal association cortex of mice injected with 100 µg of PEI-MSNs in the caudal vein (**A**) (*n* = 5), or with 3 dosages of 35 µg of PEI-MSNs given intranasally (**B**) (*n* = 5). After 8 h, PEI-MSNs were found associated with the brain endothelium (**A**, arrow right panel) and in the brain parenchyma. (**C**) Confocal images of the PEI-MSNs (arrows, red) in the hippocampus area of IV-injected (left) or intranasally administered mice (right). (**D**) Confocal images of the PEI-MSNs (arrows, red) in the olfactory bulb granular layer of IV-injected (left) or intranasally administered mice (right). (**E**) Confocal images of the PEI-MSNs (arrows, red) IV-injected (left) or intranasally administered (right) to mice, xenografted with BT-12 cells. Cell nuclei are counterstained with DAPI. (**F**) Confocal images of the PEI-MSNs (arrows, red) IV-injected (left) or intranasally administered (right) to mice, xenografted with U-87MG-GFP cells. Cell nuclei are counterstained with DAPI and blood vessels labeled with anti-CD31 (green).

## Data Availability

The data presented in this study are available in this article (and Appendix A).

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
