# Peer review of "Circumventing Drug Treatment? Intrinsic Lethal Effects of Polyethyleneimine (PEI)-Functionalized Nanoparticles on Glioblastoma Cells Cultured in Stem Cell Conditions"

_cancers, 2021, doi:10.3390/cancers13112631_

Round 1

Reviewer 1 Report

The authors have not truly resolved the issue regarding the culturing of cells using the standard vs. stem cell conditions. Regardless, to clarify for the readers, please specify in the figure legends which media conditions were used for each cell line, particularly for figure 1 which has data for both T98G under standard conditions and BT-3-CD133+, BT-12, and BT13 in stem cell conditions so that readers that may only look at the figures will appreciate the difference. 

Author Response

We are grateful for the opportunity to resubmit our study revised based on reviewer´s constructive comments. As listed below, we have addressed comprehensively all the reviewer´s questions and requests. We have also modified the manuscript to very clearly emphasize also the limitations of the data. Based on absolute novelty of the data, and its potential future implications, we sincerely hope that these revisions now allows acceptance of the submitted work for publication in Cancers.  

Reviewer 1:

  1. The authors have not truly resolved the issue regarding the culturing of cells using the standard vs. stem cell conditions. Regardless, to clarify for the readers, please specify in the figure legends which media conditions were used for each cell line, particularly for figure 1 which has data for both T98G under standard conditions and BT-3-CD133+, BT-12, and BT13 in stem cell conditions so that readers that may only look at the figures will appreciate the difference.

Authors response: We appreciate this very constructive and useful suggestion and we now have added either “cultured in stem cell conditions” or “cultured in standard conditions” to each figure legend where relevant.

Reviewer 2 Report

In the article by Prabhakar et al., the authors study the effect of polyethylenimine-functionalized nanoparticles on glioma cell lines and patient-derived glioblastoma stem cells. A large number of experiments have been carried out to support their ideas, however there is an impression that the conclusions are built mainly on hypotheses and some important controls are missing. Please consider the following suggestions and think about possible solutions how to improve your article and present your work in the way it gives an impression of undoubtful proof of the conclusions you make.

  1. The Introduction section is overloaded with the literature review to some extent. It would be favorable to shrink and clarify it by leaving only essential statements to introduce your research. Most of the second paragraph [Conventional targeting….. antagonist [15].], for example, could be safely moved to the Discussion. Please state the hypothesis and the objective(s) without disclosing all the results, discussing them, and making conclusions still in the Introduction section.
  2. Please add characteristics to the patient-derived cell lines (MIB-1 index, methylation status, IDH, etc.) The more characteristics you give the clearer the understanding can be. The link to the reference [7] leads to the review without clear explanation of the cultures you are using.
  3. It is generally known that the established cell lines can be cultivated and reseeded almost infinitely, and therefore they are “established”. Patient-derived cell lines might be more vulnerable to the culturing conditions and might have the limited number of passages. Of course, the authors used stem cells (though their characteristics are still unclear), that can be potentially reseeded for a long time, however there is no information on the passages the authors have used, and there might be still a reseeding limit that could influence the results of the experiments.
  4. No proper control by TEM imaging of any of the established glioma cells lines is provided. What if your observations are the same for BT-12 and for T98G cells, for example?
  5. In the in vivo procedures, please clarify the number of animals you have used in the experiments.
  6. The last paragraph in the Results section is the discussion on the intranasal delivery (lines 504-510). Please move the discussion-containing paragraphs to Discussion.
  7. As stated by the authors, PEI itself is initially toxic for cells. Could the authors make a proper control of cell incubation with PEI even in reasonably small concentrations? Could the authors compare the toxicity and concentrations of the PEI in MSNs and free PEI? Could there be a possible link to PEI toxicity when the nanoparticles are released inside tumor cells?
  8. Please check the manuscript for English language mistakes/misprints:

Line 28: “serum containing medium”, Line 512: “PEI functionalized”, Line 517:,”polyethyleneimine PEI”, for example, etc.

  1. There is a general impression that the authors build their conclusion around one experiment on cytotoxicity, and even that is lacking some important controls.
  2. Was the animal study approved by any ethic committee? 

Author Response

We are grateful for the opportunity to resubmit our study revised based on reviewer´s constructive comments. As listed below, we have addressed comprehensively all the reviewer´s questions and requests. We have also modified the manuscript to very clearly emphasize also the limitations of the data. Based on absolute novelty of the data, and its potential future implications, we sincerely hope that these revisions now allows acceptance of the submitted work for publication in Cancers.  

Reviewer 2:

  1. The Introduction section is overloaded with the literature review to some extent. It would be favorable to shrink and clarify it by leaving only essential statements to introduce your research. Most of the second paragraph [Conventional targeting….. antagonist [15].], for example, could be safely moved to the Discussion. Please state the hypothesis and the objective(s) without disclosing all the results, discussing them, and making conclusions still in the Introduction section.

Authors response: We agree with the reviewer´s comment and have now overall shortened the introduction. We also removed most of the suggested paragraph “[Conventional targeting….. antagonist [15]”. Discussion in the hypothesis and objectives paragraph has also been removed.

  1. Please add characteristics to the patient-derived cell lines (MIB-1 index, methylation status, IDH, etc.) The more characteristics you give the clearer the understanding can be. The link to the reference [7] leads to the review without clear explanation of the cultures you are using.

Authors response: We are very sorry for inclusion of a wrong reference.  The reference [7] has now been removed and we cite to our previous original publication in which the cells have been described [Ref. 30; Le Joncour et al 2019] (line 138). These GSCs have been previously analyzed by RNA sequencing and we now provide Gene Expression Omnibus submission reference (GSE169418) for a reader to access the data (line 140). We also included the information that according to the molecular classification of glioblastomas the cells belong to the Mesenchymal subtype. An even more detailed description of these GSCs will be published in the near future (manuscript in press) but unfortunately these data is under press embargo until publication.

  1. It is generally known that the established cell lines can be cultivated and reseeded almost infinitely, and therefore they are “established”. Patient-derived cell lines might be more vulnerable to the culturing conditions and might have the limited number of passages. Of course, the authors used stem cells (though their characteristics are still unclear), that can be potentially reseeded for a long time, however there is no information on the passages the authors have used, and there might be still a reseeding limit that could influence the results of the experiments.

Authors response: In these experiments the GSCs had a passage number of 25-30. In other studies the same cells have been also cultured almost up to 40 passages and we have not observed obvious changes in their behavior, or vulnerability of cells due to passaging.

  1. No proper control by TEM imaging of any of the established glioma cells lines is provided. What if your observations are the same for BT-12 and for T98G cells, for example?

Authors response: In these TEM experiments BT-12 cells untreated with PEI-MSNs were used as controls and this control data is now displayed as figure 8. In addition, we now show data that in MDA-MB-231 cells treated with PEI-MSNs, the particles were localized in the vesicular space and no visible damage to cellular ultrastructure was observed (Figure S7). Therefore, also at the level of subcellular organelle structure, the BT-12 cells show clearly different response than established cancer cells represented here by MDA-MB-231.

  1. In the in vivo procedures, please clarify the number of animals you have used in the experiments.

Authors response: We have now clarified the number of animals we used on the experiments. Information can be found in methods on line 268 and in the figure 10 legend.

  1. The last paragraph in the Results section is the discussion on the intranasal delivery (lines 504-510). Please move the discussion-containing paragraphs to Discussion.

Authors response: As suggested, the text in lines 504-510 have been moved to discussion (lines 554-559).

  1. As stated by the authors, PEI itself is initially toxic for cells. Could the authors make a proper control of cell incubation with PEI even in reasonably small concentrations? Could the authors compare the toxicity and concentrations of the PEI in MSNs and free PEI? Could there be a possible link to PEI toxicity when the nanoparticles are released inside tumor cells?

Authors response: The reviewer has a very valuable point as cytotoxicity of free PEI is indeed a very well-known property.  As now stated in the introduction (lines 67-69), we have previously compared the cytotoxicity between free PEI and PEI-MSNs previously in HeLa cells and incorporation of PEI to MSN nanoparticles dramatically decreased the cytotoxicity [ref. 23 in our current manuscript; Desai et al.  Mesoporous Biomater. 2014, 1, 16–43, doi: 10.2478/mesbi-2014-0001 (Fig. 15)]. These results are now repeated in the current manuscript where PEI-MSNs indeed were non-toxic to established cancer cells. Therefore, the selective toxicicity of PEI-MSNs in GSCs instead was such a great surprise.  This surprising vulnerability of GSCs to PEI-MSNs, first time reported here, does constitute the important novelty of our study and regardless of some potential limitations of our data, we strongly feel that this finding may have important ramifications to at least GSC biology and therapy.

  1. Please check the manuscript for English language mistakes/misprints:
    1. Line 28: “serum containing medium”, Line 512: “PEI functionalized”, Line 517: ”polyethyleneimine PEI”, for example, etc.

Authors response: These mistakes have been corrected and we have thoroughly checked the manuscript for language errors.

  1. There is a general impression that the authors build their conclusion around one experiment on cytotoxicity, and even that is lacking some important controls

Authors response: Although the original discovery of the cytotoxicity of PEI-MSN towards GSCs in stem cell culture conditions was indeed based on the single experiment this observation has been repeated in multiple experiments across the study: Fig. 1 and S3 (colony growth), Fig. 4 (Western blot), Fig. 6, 7 and S8 (Transmission electron microscopy), Fig. 9 (in vitro blood-brain barrier). Therefore, we are very confident of the reproducibility of the reported phenomenon.

  1. Was the animal study approved by any ethic committee?

Authors response: The information has been added to line 265-267: “All experiments involving animals were authorized by the National Animal Experimental Board in Finland (Helsinki, Finland), under the licenses 

Round 2

Reviewer 1 Report

Thank you for making the changes.

Author Response

Circumventing drug treatment? Intrinsic lethal effects of polyethyleneimine (PEI) functionalized nanoparticles on glioblastoma cells cultured in stem cell conditions

Neeraj Prabhakar 1,†, Joni Merisaari 2,3,†, Vadim Le Joncour 4, Markus Peurla 3, Didem Åžen Karaman 1,5, Eudald Casals 1,6, Pirjo Laakkonen 4,7, Jukka Westermarck 2,3,* and Jessica M. Rosenholm1,* 

Authors response to reviewer´s comments

We are grateful for the opportunity to resubmit our study revised based on reviewer´s constructive comments. As listed below, we have addressed comprehensively all the editor´s questions and requests based on reviewer´s comments. Based on absolute novelty of the data, and its potential future implications, we sincerely hope that these revisions now allow acceptance of the submitted work for publication in Cancers.  

1) I suggest the authors to show a graphic schematic of PEI-MSNs to help the reader understand the molecule better.

Thank you for the suggestion. We have added a schematic presentation of PEI-MSN as figure 1. All other figure labels have been adjusted.

2) Discussion. The part that covers lines 513 to 519 is irrelevant as these aspects of GSCs have not been studied here. I recommend deletion.

This has been deleted as suggested.

3) Discussion. Lines 522-523. “considering that GB cells grow in very heterogenous conditions in tumors” lacks evidence and should be deleted. “under selective growth conditions” should be deleted as well.

This has been deleted as suggested.

4) Figure S5. This should have used BT-12 cells instead of BT-3 cells since the authors said that “we selected the BT-12 GSCs for an in-depth analysis” (page 8). Flow cytometry needs to quantify the fractions of each quadrant.

Data about each quadrant has been added to the figure S5. While we agree that BT-12 cells would have been better to be used for the flow cytometry experiment these cultures got contaminated just before the experiment so we had to use BT-3 which we had in culture at the time. Although mix of the cell lines could be considered as a caveat, it on the other hand shows that the lack of effects of MSN-PEI on apoptosis can be generalized across the cell lines.

5) Figure 8 is just a control and does not deserve an independent figure. Should be combined with Figure 7.

Thank you for the suggestion. The figures have been combined and figure legend has been adjusted (lines 396-406)

6) Figure 5 showed green mitochondria stain and red PEI-MSNs, but in the Methods section, the indicated colors were the opposite. Something is not right.

The statement is correct. We have used the pseudo-coloring to match with the existing colors of PEI-MSNs. The actual emission of Mitochondrial stain is orange/ red and MSN-PEI is green due to FITC.

“FITC-conjugated PEI-MSNs were excited with 488 nm argon laser and emission was collected by green channel (510-550 nm). The Mitotracker Orange® was excited by 561 nm laser and emission were collected at 575-610 nm.” 

7) In the BBTB assay in vitro, what media was used in the lower chamber that had BT-12 cells. It was said BT-12 gliospheres so I assume it was stem cell media. But this has to be clarified. In Discussion, the authors said “killing activity towards GSC even under similar experimental conditions” (lines 511-512). What are “the similar experimental conditions”? Not clear.

The BBTB in vitro assay is very extensively explained in the ref [56] which actually also contains data obtained with nanoparticles, so we thought it was unnecessary to go through every step of the assay. Nevertheless, we have added some explanations, lines 248-254 and 432-445. We have also added an explanatory diagram on the Fig. S9 regarding the membrane’s pores.

8) Section 3.4. in the Results. When describing Figure 6, the authors never used the term “lysosome”, sticking to the term “vesicles”, while the section’s title, abstract and other parts used “lysosomal membrane rupture”. This feels a bit discordant. Please clarify.

We have changed the wording to lysosomes

9) In the Methods section, the authors said that they used either 24 or 72 hours for the experiments of TEM. However, in any related figures and supplemental figures, it was not indicated if 24 hours or 72 hours treatment was used.

Thank you for noting this. We have added information to each figure legend.

Figure 7 24h

Figure 8, 72h

Figure S7, 72h

Figure S8, 72h

10) Figure S9 was labeled as Figure S8. In A, it was not clear why and how one dotted circle is indicated where there should be many pores.

Figure S9 is now correctly labeled. Additionally, we have now added a schematical representation of the assay and visual indication all across the Fig S9. The legend has been completed accordingly.

11) Figure 9C right. What is the small picture on the right and the white square in it? This picture lacked scale bar.

We have slightly re-arranged the panels and added visual indications of the designated region of interest. Scalebar has been added and legend edited.

12) Line 470. I did not understand what “fenestrated BBB before the cribriform plate” means. And why does the BBB matter when the nanoparticles are delivered intranasally?

We have added anatomical description of the intranasal route to the brain starting at the line 475. We also studied the brain distribution of PEI-MSNs from the mice treated with PEI-MSNs intravenously and therefore BBB penetration properties of the particles is fully relevant to the study.

13) In the Methods section, blocking in western blot was said twice and this has to be fixed.

This mistake has been corrected and the antibody list slightly clarified on lines 172-175.

14) Line 358. Delete “S7”.

“S7” has been deleted from line 358.

15) Figure 9D. It was said that the assay was done at 72 hours after the nanoparticles treatment. This has to be mentioned in the Methods section as well.

This has been added to the methods on line 261.

16) Figure S3. Is this only a repeat (confirmation) of Figure 1A? If so, please mention that.

Similar experiment, with same settings, has been done numerous times. As at the time the result was so surprising because we thought empty PEI-MSNs should not be toxic to any cell type or in different culture conditions.

17) Figure 7. What is the rupture of the cristae? Please indicate using an arrow in the figure.

This has been clarified in the figure 7 C-D and in the figure legend (lines 398-404). We also added arrows as you suggested.

18) Line 478. The phrase here should be something like: at densities similar to what was observed in the rest of the brain.

Thank you for the suggestion. Text has been changed on line 501.

19) Line 560. What is LMP?

LMP stands for lysosomal membrane permeabilization. However, this abbreviation is not needed here. Correction has been made to line 584.

Reviewer 2 Report

Within this second revision, the authors have addressed all the reviewer’s questions and comments and have improved the manuscript significantly by modifying the introduction and discussion sections, adding more details in the materials and methods and the results, which clarified all the uncertainties. Based on the overall evaluation of the final version of the manuscript I can recommend it for publication in Cancers.

Author Response

Circumventing drug treatment? Intrinsic lethal effects of polyethyleneimine (PEI) functionalized nanoparticles on glioblastoma cells cultured in stem cell conditions

Neeraj Prabhakar 1,†, Joni Merisaari 2,3,†, Vadim Le Joncour 4, Markus Peurla 3, Didem Åžen Karaman 1,5, Eudald Casals 1,6, Pirjo Laakkonen 4,7, Jukka Westermarck 2,3,* and Jessica M. Rosenholm1,*

 Authors response to reviewer´s comments

We are grateful for the opportunity to resubmit our study revised based on reviewer´s constructive comments. As listed below, we have addressed comprehensively all the editor´s questions and requests based on reviewer´s comments. Based on absolute novelty of the data, and its potential future implications, we sincerely hope that these revisions now allow acceptance of the submitted work for publication in Cancers.  

 1) I suggest the authors to show a graphic schematic of PEI-MSNs to help the reader understand the molecule better.

Thank you for the suggestion. We have added a schematic presentation of PEI-MSN as figure 1. All other figure labels have been adjusted.

2) Discussion. The part that covers lines 513 to 519 is irrelevant as these aspects of GSCs have not been studied here. I recommend deletion.

This has been deleted as suggested.

3) Discussion. Lines 522-523. “considering that GB cells grow in very heterogenous conditions in tumors” lacks evidence and should be deleted. “under selective growth conditions” should be deleted as well.

This has been deleted as suggested.

4) Figure S5. This should have used BT-12 cells instead of BT-3 cells since the authors said that “we selected the BT-12 GSCs for an in-depth analysis” (page 8). Flow cytometry needs to quantify the fractions of each quadrant.

Data about each quadrant has been added to the figure S5. While we agree that BT-12 cells would have been better to be used for the flow cytometry experiment these cultures got contaminated just before the experiment so we had to use BT-3 which we had in culture at the time. Although mix of the cell lines could be considered as a caveat, it on the other hand shows that the lack of effects of MSN-PEI on apoptosis can be generalized across the cell lines.

5) Figure 8 is just a control and does not deserve an independent figure. Should be combined with Figure 7.

Thank you for the suggestion. The figures have been combined and figure legend has been adjusted (lines 396-406)

6) Figure 5 showed green mitochondria stain and red PEI-MSNs, but in the Methods section, the indicated colors were the opposite. Something is not right.

The statement is correct. We have used the pseudo-coloring to match with the existing colors of PEI-MSNs. The actual emission of Mitochondrial stain is orange/ red and MSN-PEI is green due to FITC.

“FITC-conjugated PEI-MSNs were excited with 488 nm argon laser and emission was collected by green channel (510-550 nm). The Mitotracker Orange® was excited by 561 nm laser and emission were collected at 575-610 nm.” 

7) In the BBTB assay in vitro, what media was used in the lower chamber that had BT-12 cells. It was said BT-12 gliospheres so I assume it was stem cell media. But this has to be clarified. In Discussion, the authors said “killing activity towards GSC even under similar experimental conditions” (lines 511-512). What are “the similar experimental conditions”? Not clear.

The BBTB in vitro assay is very extensively explained in the ref [56] which actually also contains data obtained with nanoparticles, so we thought it was unnecessary to go through every step of the assay. Nevertheless, we have added some explanations, lines 248-254 and 432-445. We have also added an explanatory diagram on the Fig. S9 regarding the membrane’s pores.

8) Section 3.4. in the Results. When describing Figure 6, the authors never used the term “lysosome”, sticking to the term “vesicles”, while the section’s title, abstract and other parts used “lysosomal membrane rupture”. This feels a bit discordant. Please clarify.

We have changed the wording to lysosomes

9) In the Methods section, the authors said that they used either 24 or 72 hours for the experiments of TEM. However, in any related figures and supplemental figures, it was not indicated if 24 hours or 72 hours treatment was used.

Thank you for noting this. We have added information to each figure legend.

Figure 7 24h

Figure 8, 72h

Figure S7, 72h

Figure S8, 72h

10) Figure S9 was labeled as Figure S8. In A, it was not clear why and how one dotted circle is indicated where there should be many pores.

Figure S9 is now correctly labeled. Additionally, we have now added a schematical representation of the assay and visual indication all across the Fig S9. The legend has been completed accordingly.

11) Figure 9C right. What is the small picture on the right and the white square in it? This picture lacked scale bar.

We have slightly re-arranged the panels and added visual indications of the designated region of interest. Scalebar has been added and legend edited.

12) Line 470. I did not understand what “fenestrated BBB before the cribriform plate” means. And why does the BBB matter when the nanoparticles are delivered intranasally?

We have added anatomical description of the intranasal route to the brain starting at the line 475. We also studied the brain distribution of PEI-MSNs from the mice treated with PEI-MSNs intravenously and therefore BBB penetration properties of the particles is fully relevant to the study.

13) In the Methods section, blocking in western blot was said twice and this has to be fixed.

This mistake has been corrected and the antibody list slightly clarified on lines 172-175.

14) Line 358. Delete “S7”.

“S7” has been deleted from line 358.

15) Figure 9D. It was said that the assay was done at 72 hours after the nanoparticles treatment. This has to be mentioned in the Methods section as well.

This has been added to the methods on line 261.

16) Figure S3. Is this only a repeat (confirmation) of Figure 1A? If so, please mention that.

Similar experiment, with same settings, has been done numerous times. As at the time the result was so surprising because we thought empty PEI-MSNs should not be toxic to any cell type or in different culture conditions.

17) Figure 7. What is the rupture of the cristae? Please indicate using an arrow in the figure.

This has been clarified in the figure 7 C-D and in the figure legend (lines 398-404). We also added arrows as you suggested.

18) Line 478. The phrase here should be something like: at densities similar to what was observed in the rest of the brain.

Thank you for the suggestion. Text has been changed on line 501.

19) Line 560. What is LMP?

LMP stands for lysosomal membrane permeabilization. However, this abbreviation is not needed here. Correction has been made to line 584.

This manuscript is a resubmission of an earlier submission. The following is a list of the peer review reports and author responses from that submission.

Round 1

Reviewer 1 Report

The authors of, "Circumventing drug treatment? Polyethylenimine functionalized
nanoparticles inherently and selectively kill patient-derived
glioblastoma stem cells" have presented a series of experiments that demonstrated that PEI nanoparticles may reduce the viability of GBM cells. Their data suggest that the response is restricted to glioma stem cells on the basis that cell viability in the colony assay is unaffected in the standard cell lines T98G, A172, U87MG, or MDA-MB-231. It would be more impactful to know if their findings from their patient-derived glioma stem cells would be recapitulated in stem cell populations from those standard cell lines if they were also grown in the no serum, FGF and EGF media. There is also some ambiguity in the methods. The authors state, "For colony growth and microscopy GSC populations were cultured as monolayers on Matrigel". What about the conventional cell lines? Were they also cultured on Matrigel for those colony growth assays? There is also no information about the culture conditions of the A172, U87MG, MDA-MB-231, or HeLa cells, also presumably they were similar to the T98G. Also, the value of Figure S3 is limited since that figure is in the body of the paper. 

The Western blot method would also benefit from some clarification. In figure 2, all of those targets were clearly blotted from the same gel and blot. Was the membrane cut, or was the membrane stripped and reprobed? What method of stripping was performed, and how was it verified that the signal obtained was a consequence of the intended antibody and not carry-over? 

Author Response

Response to reviewers:

 We were very pleased to find out that all reviewers clearly appreciated the novelty and importance of our original findings, and provided very useful and constructive criticism to further improve the study. As listed in the attached detailed response to the reviewers, we have addressed comprehensively reviewer´s questions and requests. Based on absolute novelty of the data, and its potential future implications, we sincerely hope that the provided additional data now allows acceptance of the submitted work for publication in Cancers.  

Reviewer 1:

The authors of, "Circumventing drug treatment? Polyethylenimine functionalized nanoparticles inherently and selectively kill patient-derived glioblastoma stem cells" have presented a series of experiments that demonstrated that PEI nanoparticles may reduce the viability of GBM cells. Their data suggest that the response is restricted to glioma stem cells on the basis that cell viability in the colony assay is unaffected in the standard cell lines T98G, A172, U87MG, or MDA-MB-231. It would be more impactful to know if their findings from their patient-derived glioma stem cells would be recapitulated in stem cell populations from those standard cell lines if they were also grown in the no serum, FGF and EGF media.

A: Thank you for a very valid point. We agree that we do not know how different culture condition would affect the cell viability of stem cell population from other standard cells lines. Furthermore, as correctly stated by the reviewer, another limitation arises from different cell culture methods of established GB cell lines and GSCs. Thus, it would be in the future important to investigate if this phenomenon validates throughout different stem cells populations within the tumor microenvironment, consisting of the tumor cells and the stem cells of the same origin. During the study, we actually tried to culture the T98G, A172 and U87MG cells in the same environment as the GSCs, but they did not grow or had very limited growth in the absence of serum. Very important additional evidence for the non-toxicity of the used nanoparticles for normal cells, and their GSC selective effect, was that in the in vitro BBB model, the viability of endothelial cells and astrocytes was not significantly affected compared to the untreated cells, whereas nanoparticles significantly affected the viability of BT-12 GSCs (Fig. 6D). We have now mentioned these limitations in the discussion section (pg. 14, 1st chapter). We have also modified the title of the manuscript by removing any mention about selectivity of PEI-MSNs towards GSCs.

There is also some ambiguity in the methods. The authors state, "For colony growth and microscopy GSC populations were cultured as monolayers on Matrigel". What about the conventional cell lines? Were they also cultured on Matrigel for those colony growth assays?

A: GSCs need to be cultured on Matrigel to ensure that they grow as a monolayer under our experimental design; otherwise, they will grow as spheroids. Established cell lines were not cultured on Matrigel as it is not needed to culture them as monolayer. We have now mentioned these limitations in the discussion section (pg. 14, 1st chapter).

There is also no information about the culture conditions of the A172, U87MG, MDA-MB-231, or HeLa cells, also presumably they were similar to the T98G. Also, the value of Figure S3 is limited since that figure is in the body of the paper. 

A: We have now added this information in the methods section. Figure S3 is from a different batch of nanoparticles. Now this information has been added to the supplementary information.

The Western blot method would also benefit from some clarification. In figure 2, all of those targets were clearly blotted from the same gel and blot. Was the membrane cut, or was the membrane stripped and reprobed? What method of stripping was performed, and how was it verified that the signal obtained was a consequence of the intended antibody and not carry-over? 

A: Please see the revised text in the methods section.  Apoptotic and autophagic proteins were blotted from the same membrane to ensure similar settings. Therefore, first the apoptotic proteins PARP-1, cPARP (and loading control β-actin) we blotted. After this, the membrane was stripped and the autophagic proteins P62 and LC3-β were reblotted on the same membrane. There is no chance of carry-over as the molecular weights of the proteins are not similar. Furthermore, none of the bands are “identical” hence, they should not be the same.

PARP-1 - 116 kDa (Rabbit)

cPARP – 25, 89 kDa (Rabbit)

P62 - 65 kDa (Mouse)

LC3-β - 14 kDa (Rabbit)

β-actin - 43 kDa (Mouse)

Reviewer 2 Report

Reviewer’s comments

The article by Prabhakar et al. describes the effect of polyethylenimine-functionalized nanoparticles on glioma cell lines and patient-derived glioblastoma stem cells. A series of extensive experiments have been carried out by the authors of the manuscript to support their ideas. As malignant gliomas and particularly glioblastoma multiforme (GBM) remain uncurable regardless of any modern therapies, and radiation and temozolomide can slightly prolong patient overall survival, any experiments that can support further therapy development deserve publishing. However, it is difficult to accept the manuscript in the present form, as the following issues need to be clarified by the authors before further evaluation:

  1. Please check the Author’s instructions and already published articles to rewrite your manuscript in the proper way. The Introduction contains literature review that could be better placed in Discussion and conclusions that should be stated in the end of the article in the Conclusion or Discussion section. The Discussion with the proper literature review and comparative analysis of other related papers is almost missing. The limitations of the study are not disclosed as they should be in relation to the controls in this manuscript.
  2. The novelty of the study should be more clearly explained in the Introduction.
  3. The title looks overly flashy showing the “selectivity” of the nanoparticles towards glioma stem cells. The selectivity of the compound to kill only tumor stem cells can be verified by its actual selective influence on the stem cells within the tumor microenvironment, consisting of the tumor cells and the stem cells of the same origin. The authors tried show the selective killing effect of the developed nanoparticles on patient-derived glioma stem cells using well-established glioma cell lines as controls. In this regard, though other options are limited, the control cannot absolutely meet the above selectivity parameters. As well-established cell lines are designed for infinite division and growth, and naturally can be more resistant to various kinds of influences compared to patient-derived cells.
  4. Regarding nanoparticles, depending on the size and the surface structure and charge they can be toxic to some cells and non-toxic to others, and this can be seen in case of gold nanoparticles, for example, as they were extensively studied with numerous results published in the literature. Though close to real conditions, the experiments shown in this article still remain synthetic, and all the related issues should be described and discussed in the limitations of the study.
  5. Could the porous structure of the particles contain remnants of the substances or solvents used during their production or further processing that could influence the cell survival?
  6. The materials should be properly described showing the company, the city (the state), and the country when first mentioned, there is no need to repeat the origin details further in the text.
  7. In the methods, the selected way to fix and stain the cells might bring additional staining of unrelated to cells particles, that can be seen in Figure 1. I would recommend to use glutaraldehyde for colonies fixation. (Franken, N., Rodermond, H., Stap, J. et al. Clonogenic assay of cells in vitro. Nat Protoc 1, 2315–2319 (2006). https://doi.org/10.1038/nprot.2006.339)
  8. Figure S5 could better fit the main body of the article, as it carries valuable information for the proof of the main hypothesis.
  9. Please proofread the article for English grammar mistakes and misprints, such as: Title: “Polyethylenimine functionalized”; Page 4: “Antibodies……was….; antibodies ….was”; Page 5: “…coated glass coverslips glass coverslips and…”, etc.

To conclude, the authors have conducted a very interesting and valuable study, and to be published they need to modify the article to make it clearer to the readers.

Author Response

Response to reviewers:

 We were very pleased to find out that all reviewers clearly appreciated the novelty and importance of our original findings, and provided very useful and constructive criticism to further improve the study. As listed in the attached detailed response to the reviewers, we have addressed comprehensively reviewer´s questions and requests. Based on absolute novelty of the data, and its potential future implications, we sincerely hope that the provided additional data now allows acceptance of the submitted work for publication in Cancers.  

Reviewer 2

The article by Prabhakar et al. describes the effect of polyethylenimine-functionalized nanoparticles on glioma cell lines and patient-derived glioblastoma stem cells. A series of extensive experiments have been carried out by the authors of the manuscript to support their ideas. As malignant gliomas and particularly glioblastoma multiforme (GBM) remain uncurable regardless of any modern therapies, and radiation and temozolomide can slightly prolong patient overall survival, any experiments that can support further therapy development deserve publishing. However, it is difficult to accept the manuscript in the present form, as the following issues need to be clarified by the authors before further evaluation:

  1. Please check the Author’s instructions and already published articles to rewrite your manuscript in the proper way. The Introduction contains literature review that could be better placed in Discussion and conclusions that should be stated in the end of the article in the Conclusion or Discussion section. The Discussion with the proper literature review and comparative analysis of other related papers is almost missing. The limitations of the study are not disclosed as they should be in relation to the controls in this manuscript.

A: We thank the reviewer for this suggestion. Please see the revised version of the manuscript. We have tried our best to include the most relevant literature in the discussion section. The limitation of our study is now also mentioned in the revised manuscript (pg. 14 and 15).

  1. The novelty of the study should be more clearly explained in the Introduction.

A: We intentionally tried to avoid direct statements related to “novel” or “unprecedented” findings as we feel they are not appropriate to be used in the scientific articles. As for now, how we have tried to structure the text so that every section in the Introduction outlines what is state-of-the-art of what is being discussed in that section, ultimately leading to the final sentence “this discovery of the inherent role of PEI-MSNs in selectively eradicating otherwise highly resistant GSCs presents a novel vulnerability to exploit for brain cancer (GB) treatment.” The word “discovery” here is meant to convey the most important novelty aspect of the study. In addition, the new Discussion part (pages 14-15) should now hopefully also shed more light on the novelty of our findings.

  1. The title looks overly flashy showing the “selectivity” of the nanoparticles towards glioma stem cells. The selectivity of the compound to kill only tumor stem cells can be verified by its actual selective influence on the stem cells within the tumor microenvironment, consisting of the tumor cells and the stem cells of the same origin. The authors tried show the selective killing effect of the developed nanoparticles on patient-derived glioma stem cells using well-established glioma cell lines as controls. In this regard, though other options are limited, the control cannot absolutely meet the above selectivity parameters. As well-established cell lines are designed for infinite division and growth, and naturally can be more resistant to various kinds of influences compared to patient-derived cells.

A: We agree with this notion and have modified the title of the manuscript by removing any mention about selectivity of PEI-MSNs towards GSCs. It is true that we do not know how the different culture conditions would affect the cell viability of stem cell population from other standard cells lines and agree that in future it would be important to investigate if this phenomenon validates throughout different stem cells populations within the tumor microenvironment, consisting of the tumor cells and the stem cells of the same origin. This is now mentioned in the discussion section of revised manuscript as a limitation of our study (pg. 15). Very important additional evidence for the non-toxicity of the used nanoparticles for normal cells was that in the in vitro BBB model, the viability of endothelial cells and astrocytes was not significantly affected compared to the untreated cells whereas nanoparticles significantly affected the viability of the BT-12 GSCs (Fig. 6D). This is also now mentioned in the discussion (Pg. 14)

  1. Regarding nanoparticles, depending on the size and the surface structure and charge they can be toxic to some cells and non-toxic to others, and this can be seen in case of gold nanoparticles, for example, as they were extensively studied with numerous results published in the literature. Though close to real conditions, the experiments shown in this article still remain synthetic, and all the related issues should be described and discussed in the limitations of the study.

A: Gold nanoparticles (non-biodegradable) and our bioerodible mesoporous silica nanoparticles are two vastly diverse types of nanomaterials so making that direct comparison may not be relevant in our case. However, we totally agree and it is also quite well-known that physicochemical characteristics and especially size and surface functionalization (charge) could affect cell viability as a function of cell type; which is also the very outcome of our study. Our results show that that PEI (surface functionalization), but not the nanoparticles per se, is crucial for inducing the proton-sponge effect to permeabilize the vulnerable lysosomes of GSCs. The effect is neither observed for the same particles without PEI surface functionalization (Fig. S5) nor for GB cells without GSC characteristics. This crucial aspect is discussed in both the Introduction section as well as in the Discussion of the revised manuscript (pg.14-15).

  1. Could the porous structure of the particles contain remnants of the substances or solvents used during their production or further processing that could influence the cell survival?

A: This is a very valid question since the particle synthesis process certainly involves a known cytotoxic substance (the pore templating agent – a cationic surfactant) but in this case, we’d like to say it is highly unlikely. Should this be the case, the highest risk for encountering this scenario would be for the non-functionalized particles and for those, we did not detect any cytotoxic effect (Fig. S5).  Also, we observed identical results with two independent patches of nanoparticles (Fig. S3). Therefore, we find it very unlikely that GCS selective cell killing by PEI would be caused by any impurities from the nanoparticle production.

  1. The materials should be properly described showing the company, the city (the state), and the country when first mentioned, there is no need to repeat the origin details further in the text.

A: Thank you for the comment. This has been now added to the revised manuscript. Please see the updated text. 

  1. In the methods, the selected way to fix and stain the cells might bring additional staining of unrelated to cells particles, that can be seen in Figure 1. I would recommend to use glutaraldehyde for colonies fixation. (Franken, N., Rodermond, H., Stap, J. et al. Clonogenic assay of cells in vitroNat Protoc 1, 2315–2319 (2006). https://doi.org/10.1038/nprot.2006.339)

A: We thank the reviewer for this information. We will keep a note of it. We typically use PFA for colony fixation. Based on our experience the most unbound crystal violet will be removed upon multiple washing steps.

  1. Figure S5 could better fit the main body of the article, as it carries valuable information for the proof of the main hypothesis.

A:The figure is now changed to S4.Yes, we agree that figure is important for the hypothesis. However, we still think that figure S4 is better suited only as a supplementary information rather than in the main manuscript because T98G in Fig 1 already serves the same purpose for comparison with established GB cell line. The readers can still easily access the information from the supplementary information.

  1. Please proofread the article for English grammar mistakes and misprints, such as: Title: “Polyethylenimine functionalized”; Page 4: “Antibodies……was….; antibodies ….was”; Page 5: “…coated glass coverslips glass coverslips and…”, etc.

A: We thank the reviewer for pointing this out and are sorry for the earlier omission on this issue. Please see the updated text.

To conclude, the authors have conducted a very interesting and valuable study, and to be published they need to modify the article to make it clearer to the readers.

A: We thank the reviewer for this positive note and hope we have now, with the conducted revision, been able to clarify our study to a degree suitable for publication.

Round 2

Reviewer 1 Report

The authors have submitted a revised manuscript in which they removed the word "selective" from the title, but not from the discussion. They have added a statement to the discussion that their culture conditions were not the same for the patient-derived GBM stem cells and the established GBM cell lines, however they maintain the use of the term 'selective' based upon the blood-brain-tumor-barrier cultures, because the nanoparticles did not kill the endothelial cells and the astrocytes that were part of the culture system. It should be noted that those cells were immortalized cells, artificially manipulated for maintained growth of the cells in culture, and as such are not representative. 

Furthermore the authors wrote in their comments, "During the study, we actually tried to culture the T98G, A172 and U87MG cells in the same environment as the GSCs, but they did not grow or had very limited growth in the absence of serum." These cells lines have a long publication history of being grown in the stem-cell inducing conditions as these authors used for their patient glioma cells. (PMIDs 22922964 and 25121761 to name two) The authors should have been more concerned about these cells failing to grow under these conditions. 

Also, the authors listed in their response that the there was no overlap of molecular weights of the targets in their Western blots, but they have identified the cleaved PARP and actin both at 25kDa. The authors should submit the unedited Western blot images with the supplemental data, and include the information about the stripping and reprobing in the manuscript. 

It should also be noted that in the revised manuscript, only some of the edits were highlighted as changes, but most were not, making the second review of the manuscript challenging. 

Reviewer 2 Report

The authors have answered the reviewer’s questions; however, some minor issues need to be corrected/verified before article publication.

  1. Please correct the origins of the materials in Materials and Methods (company, city, state, country) for all the materials. Line 116: “Alfa Aesar (Ward Hill, United States)” to Alfa Aesar (Ward Hill, MA, United States), line 119: “(St. Louis, United States)” to “(St. Louis, MO, United States)”, for example, etc.  Also, you don’t need to mention that information every time you mention a chemical compound or a medium. Please, state just once when you first mention the compound.
  2. Please check the manuscript for English language mistakes/misprints: Line 158: “gift” – “a gift”, Line 519: “data confirms” – “data confirm”, for example, etc.
  3. Please change the title in the Supplementary materials.
  4. If you don’t use plain PEI (without MSNs) as a control, justify it in the Discussion, or put in limitations.

Author Response

  1. Please correct the origins of the materials in Materials and Methods (company, city, state, country) for all the materials. Line 116: “Alfa Aesar (Ward Hill, United States)” to Alfa Aesar (Ward Hill, MA, United States), line 119: “(St. Louis, United States)” to “(St. Louis, MO, United States)”, for example, etc.  Also, you don’t need to mention that information every time you mention a chemical compound or a medium. Please, state just once when you first mention the compound.

A: Thank you for notifying us on this - these have now been fixed.

  1. Please check the manuscript for English language mistakes/misprints: Line 158: “gift” – “a gift”, Line 519: “data confirms” – “data confirm”, for example, etc.

A: The manuscript text has been carefully read once more to try to detect all language mistakes. Thank you for pointing this out and providing the specific examples above.

  1. Please change the title in the Supplementary materials.

A: Supplementary materials title has now been changed.

  1. If you don’t use plain PEI (without MSNs) as a control, justify it in the Discussion, or put in limitations.

A: Thank you for pointing this out; this is indeed a very valid point to be touched upon in the manuscript. Plain PEI cannot perhaps be regarded a direct “control” in this sense given that free polymer and PEI as a construct in a nanosystem are so vastly different with regard to their physical state (one is freely available in solution while the other is part of matter). Therefore, we may have intuitively -but wrongly-regarded this self-evident enough not to be specifically mentioned, but we fully agree that this notion obviously warrants proper addressing in our manuscript; especially given that not all readers may be that familiar with PEI and its uses. Thus, we have now included some more discussion on this in the Discussion section as suggested, and also in the Introduction clarified the distinction between our case and plain PEI (the cytotoxic properties of which is already well-known, as also mentioned in our manuscript e.g. on page 7: “well-reported fact that PEI can induce non-specific toxicity to cells if applied at higher concentrations[41–45]”). Thank you again very much for pointing this out!